# TRPM7-Mediated Ca^2+^ Regulates Mussel Settlement through the CaMKKβ-AMPK-SGF1 Pathway

**DOI:** 10.3390/ijms24065399

**Published:** 2023-03-11

**Authors:** Jian He, Peng Wang, Zhixuan Wang, Danqing Feng, Dun Zhang

**Affiliations:** 1Key Laboratory of Marine Environmental Corrosion and Bio-Fouling, Institute of Oceanology, Chinese Academy of Sciences, Qingdao 266071, China; 2Open Studio for Marine Corrosion and Protection, Pilot National Laboratory for Marine Science and Technology (Qingdao), Qingdao 266237, China; 3State-Province Joint Engineering Laboratory of Marine Bioproducts and Technology, College of Ocean & Earth Sciences, Xiamen University, Xiamen 361102, China

**Keywords:** biofouling, larval settlement, TRP channel, mechanosensing, AMPK signaling

## Abstract

Many marine invertebrates have planktonic larval and benthic juvenile/adult stages. When the planktonic larvae are fully developed, they must find a favorable site to settle and metamorphose into benthic juveniles. This transition from a planktonic to a benthic mode of life is a complex behavioral process involving substrate searching and exploration. Although the mechanosensitive receptor in the tactile sensor has been implicated in sensing and responding to surfaces of the substrates, few have been unambiguously identified. Recently, we identified that the mechanosensitive transient receptor potential melastatin-subfamily member 7 (TRPM7) channel, highly expressed in the larval foot of the mussel *Mytilospsis sallei*, was involved in substrate exploration for settlement. Here, we show that the TRPM7-mediated Ca^2+^ signal was involved in triggering the larval settlement of *M. sallei* through the calmodulin-dependent protein kinase kinase β/AMP-activated protein kinase/silk gland factor 1 (CaMKKβ-AMPK-SGF1) pathway. It was found that *M. sallei* larvae preferred the stiff surfaces for settlement, on which *TRPM7*, *CaMKKβ*, *AMPK*, and *SGF1* were highly expressed. These findings will help us to better understand the molecular mechanisms of larval settlement in marine invertebrates, and will provide insights into the potential targets for developing environmentally friendly antifouling coatings for fouling organisms.

## 1. Introduction

Many marine invertebrates have a biphasic life cycle, consisting of a planktonic larval stage and a benthic juvenile/adult stage [1]. When the planktonic larvae are fully developed, they must find an appropriate substrate to attach and transform into morphologically and physiologically distinct juveniles, rather than gather energy growing quickly or slowly into adult organisms [2]. Larval attachment and metamorphosis (both processes are referred to, in the present paper, as ‘settlement’) play an important role in the history of marine invertebrates, as it underpins the propagation and persistence of the population in the marine ecosystem [3]. Moreover, larval settlement is necessary for the supply of products for aquaculture industries worldwide, and is responsible for severe economic losses per year due to the biofouling of marine industries and maritime activities [4]. However, despite the fundamental importance of larval settlement, our understanding of the underlying molecular mechanisms is limited.

Larval settlement is a complex behavioral process involving substrate searching and exploration. During the exploration, the mechanosensitive receptor in the tactile sensor enables the organisms to sense and respond to surfaces of substratum, which converts extracellular physical stimuli to intracellular electrochemical signals [5,6]. If the physical stimuli are appropriately sensed by the mechanosensitive receptor, the individuals will secrete the adhesives for attachment, and metamorphose into benthic juveniles [6]. Thus, understanding the mechanosensitive receptor that enables the organisms to sense and respond to surfaces is a key to open the door of larval settlement in marine invertebrates.

Among the proteins required to mediate the sensation of and response to mechanical forces, mechanosensitive transient receptor potential (TRP) channels are suggested to be directly activated by stresses, converting physical stimuli to electrical signals at the cellular level [7]. The discoveries of these receptors have profoundly changed our view of how organisms sense the world. Although TRP channels have been extensively studied in model organisms (such as the nematode *Caenorhabditis elegans*, and the fruit fly *Drosophila melanogaster*) in the past decade [7], little has been reported for these channels in marine invertebrates. Recently, more and more studies have indicated that TRP channels are present in the sense organs responsible for settlement in marine invertebrates [8,9,10], however, which TRP channel in particular is involved in mechanosensing for larval settlement and how it functions is unclear.

Dreissenid mussels are well-known invasive species and economic pests in aquatic ecosystems [11,12]. These include the zebra mussel *Dreissena polymorpha*, the quagga mussel *Dreissena rostriformis bugensis* in North America and Europe [13], *Mytilopsis leucophaeata* in Europe [14], and *Mytilopsis sallei* in Australasia, East Asia, and India [15,16]. They can settle gregariously on submerged manmade structures, causing serious biofouling problems. Recently, we identified that the mechanosensitive transient receptor potential melastatin-subfamily member 7 (TRPM7) channel, highly expressed in the larval foot of the dreissenid mussel *M*. *sallei*, was involved in substrate exploration for settlement (unpublished data). However, how the TRPM7 channel triggers the larval settlement of *M. sallei* is unclear.

In this study, with electrophysiological detection, transcriptomic analysis, genes, and protein expression analyses and the yeast two-hybrid technique, we identified that the TRPM7-mediated Ca^2+^ signal was involved in triggering the larval settlement of *M. sallei* through the calmodulin-dependent protein kinase kinase β/AMP-activated protein kinase/silk gland factor 1 (CaMKKβ-AMPK-SGF1) pathway. Moreover, the larval responses to different substrates including the larval settlement preferences and gene expression changes of the involved signal transduction pathway were investigated. This work will deepen our understanding of the molecular mechanisms of larval settlement in marine invertebrates, and will provide insights into the potential targets for developing environmentally friendly antifouling coatings for fouling organisms.

## 2. Results and Discussion

### 2.1. TRPM7 Channel Triggers Larval Settlement through Ca^2+^ Signal

In our previous study (unpublished data), we proved that calcium plays an important role in the larval settlement of *M. sallei*. TRPM7, a mechanosensitive nonselective cation channel, plays a crucial role in intracellular homeostasis. To determine whether the TRPM7-mediated Ca^2+^ signal may possibly be involved in the signal transduction pathways for larval settlement in the mussel, a non-invasive micro-test technique (NMT) was employed to measure the net Ca^2+^ flux of *M. sallei* larvae in real-time (Figure 1A). There is both inflow and outflow ionic fluxion. If the outflow value is greater than the inflow value, the measured value of the Ca^2+^ flux is defined as efflux, and otherwise, it is defined as influx. A positive value of Ca^2+^ flux indicates efflux from the *M. sallei* larvae, and a negative value indicates influx. As shown in Figure 1B,C, the Ca^2+^ efflux was relatively stable (~150 pmol cm^2^ s^−2^) from the larvae in the absence of TRPM7 channel activators. When the activators were added, the Ca^2+^ efflux dramatically decreased, and even led to a Ca^2+^ influx after 10 min.

NMT is an effective tool to noninvasively acquire information on molecular/ionic fluxes for measuring real-time living samples, which has been widely used in the field of fundamental research including environmental science, life science, and agriculture [17,18]. Recently, NMT revealed that the Ca^2+^ efflux of the algae *Platymonas subcordiformis* was dramatically increased in the presence of indole derivatives, which showed that NMT could be applied for the evaluation of the antifouling performance of compounds [19]. In this study, NMT was used to determine the Ca^2+^ flux in the larvae for preliminary exploration of the possible physiological mechanism of how the TRPM7 channel regulates larval settlement. These results showed that the TRPM7 channel was responsible for mediating extracellular Ca^2+^ into cells the in *M. sallei* larvae, suggesting that the TRPM7 channel triggers larval settlement through a Ca^2+^ signal.

### 2.2. TRPM7-Mediated Ca^2+^ Signal Triggers Larval Settlement via CaMKKβ-AMPK Pathway

Recently, we have demonstrated that the AMPK plays an important role in regulating the larval settlement of *M. sallei* [20]. It is of note that AMPK activity can be regulated by intracellular Ca^2+^ concentration [21]. Moreover, with transcriptomic analysis of larval settlement [20], we confirmed that intracellular Ca^2+^ could activate AMPK through CaMKKβ based on KEGG pathway analysis (Figure 2). Therefore, it is reasonable to hypothesize that the TRPM7-mediated Ca^2+^ signal triggers larval settlement by activating the CaMKKβ-AMPK pathway.

To explore the possible action mechanism of TRPM7-mediated Ca^2+^ in triggering the larval settlement of *M. sallei*, the gene expression profiles of the TRPM7 channel CaMKKβ and AMPK in different larval developmental stages were analyzed by qRT-PCR (Figure 3A). All of the genes showed similar expression trends where the expression reached a maximum in the settled larvae, and then fell significantly in the metamorphosed (post-settled) larvae, suggesting their critical importance for larval settlement. Furthermore, to confirm that the expressions of *CaMKKβ* and *AMPK* can been influenced by the *TRPM7* channel, these gene expression changes in response to siRNA interference against *TRPM7* channel were examined. As shown in Figure 3B, the transfection of 10 μg mL^−1^ siRNA against the *TRPM7* channel including TRPM7-1 and TRPM7-2 significantly suppressed the mRNA levels of the *TRPM7* channel compared with the solvent control. Similar expression trends were also observed for *CaMKKβ* and *AMPK* where the gene expression was significantly suppressed after siRNA interference against the *TRPM7* channel. In addition, we examined the gene expression changes in response to the TRPM7 channel activators. As shown in Figure 3C, the expression of all of these genes were significantly increased after exposure to the TRPM7 channel activators.

Previous studies have revealed that Ca^2+^ could promote the larval settlement and metamorphosis of many marine invertebrate larvae including the coral *Pocillopora damicornis* [22], the ascidian *Halocynthia roretzi* [23], the barnacle *Balanus amphitrite* [24], and the abalone *Haliotis diversicolor supertexta* [25]. In addition, calmodulin (CaM), which is a Ca^2+^-binding protein regulating various downstream effectors in the cell signal pathways, was closely linked to the molecular mechanism of larval settlement and metamorphosis in the polychaete *Hydroides elegans* [26]. The expression of *CaM* and calcium homeostasis-related genes in coral *Montastraea faveolata* was also found to upregulate in the development process from planula to polyp [27]. However, the mechanistic details of how Ca^2+^ induces invertebrate larval settlement are of great importance but remain unclear.

With a combination of larval settlement experiments, morphological analyses of cells, and calcium assays, it has been reported that the sensory-secretory epithelial cells of the sponge *Amphimedon queenslandica* responded to environmental cues through calcium signaling, which triggered larval settlement via activation of nitric oxide (NO) signaling through the MAPK/ERK pathway [28]. In this study, we showed that the mRNA expression of CaMKKβ and AMPK were closely related to the TRPM7 channel, suggesting that the TRPM7-mediated Ca^2+^ signal triggers larval settlement through the CaMKKβ-AMPK pathway.

### 2.3. Yeast-Two-Hybrid Screening of AMPK-Interacting Proteins

Recently, more and more studies have indicated that AMPK signaling is vital to the larval settlement and metamorphosis in marine invertebrates including the mussel *Mytilus coruscus* [29] and the barnacle *Amphibalanus amphitrite* [30], however, the underlying molecular mechanisms are unclear. To further elucidate the biochemical mechanisms by which AMPK regulates the larval settlement of *M. sallei*, we conducted a yeast two hybrid (Y2H) assay to identify its interacting proteins. The mRNA of the cDNA Library exhibited excellent quality (Appendix A), and the average inserted fragment length of 24 selected clones (Appendix A) was above 1 kb (Appendix A). The results suggest that the cDNA library was of acceptable quality and can be adopted for further assays by the Y2H system. The pGBKT7-AMPK bait plasmid was successfully constructed as confirmed by sequencing (Appendix A).

The vectors pGBKT7-AMPK and pGBKT7 were transformed into the AH109 yeast strain, and the transformants were cultured on SD/-Trp, and SD/-Trp-X plates to detect the auto-activation activity. The colonies containing the pGBKT7 plasmid on SD/−Trp appeared, but did not on the SD/-Trp-X plates (Figure 4A) while the colonies containing the pGBKT7-AMPK plasmid on the SD/−Trp and SD/-Trp-His plates appeared, indicating that pGBKT7-AMPK had activity of auto-activation. To inhibit the auto-activation, different concentrations of 3-aminotriazole (3AT) were added to the SD/-Trp-X plates. This showed that the pGBKT7-AMPK colonies on the SD/-Trp-His plates containing 20 mM 3AT disappeared (Figure 4B), indicating that there was no activity of auto-activation. Thus, the pGBKT7-AMPK bait plasmid was qualified to be adopted in the next Y2H assay on SD/-Trp-X plates containing 20 mM 3AT.

To screen the AMPK-interacting proteins, pGBKT7-AMPK was co-transformed with the prey plasmid and planted on SD/-Trp-Leu plates, where 19 clones appeared (Figure 4C). These 19 colonies were subsequently transferred onto SD/-Trp-Leu-His + 20 mM 3AT, SD/-Trp-Leu-His-Ade + 20 mM 3AT, and SD/-Trp-Leu-His-Ade + X-α-gal + 20 mM 3AT plates. Sixteen of these 19 colonies still appeared blue, suggesting that they may be positive interactions between pGBKT7-AMPK and the screened prey plasmids (Figure 4C). The identified results of these interacting proteins is shown in Table 1.

To elucidate which AMPK-interacting protein is responsible for larval settlement in *M. sallei*, the transcriptome expression profiles of these proteins during larval settlement were analyzed. As shown in Appendix A, silk gland factor 1-like (SGF1), 6-phosphofructo-2-kinase (PFK-2), calmodulin-dependent protein kinase kinase 2 (CaMKKβ), and serine/threonine kinase 11 (STK11) were significantly highly expressed during larval settlement. Based the KEGG pathway analysis (Figure 2), it shows that CaMKKβ and STK11 can activate AMPK directly, while AMPK can activate PFK-2 by phosphorylation.

As a key regulator of energy metabolism, AMPK contributes greatly to the balance of glucose metabolism [31,32]. The regulation of glucose metabolism by AMPK mainly increases glucose uptake, promotes glycolysis, and suppresses gluconeogenesis. The larval settlement of marine invertebrates is an extremely energy-consuming process [33]. For example, proteins involved in energy metabolism processes including glycolysis have been found to be upregulated expressed during larval settlement in the barnacle *B. amphitrite* [33], and the polychaete *Pseudopolydora vexillosa* [34]. In this study, PFK-2, a key enzyme in glycolysis, was significantly highly expressed during the larval settlement of *M. sallei*, suggesting that glycolysis triggered by PFK-2 provides energy for the completion of larval settlement in the mussel.

Silk gland factor 1 (SGF1), a FOXA transcription factor that contains an evolutionarily conserved forkhead or winged-helix DNA-binding domain, finely regulates the spatial and temporal expression of a wide range of genes, which can be activated via phosphorylation [35]. It has been reported that SGF1 plays a critical role in activating fibroin gene expression in the silkworm *Bombyx mori* [36,37]. Although no study has indicated that SGF1 is vital in the larval settlement in marine invertebrates, based on the fact that the mussels settle to the substrates via the byssus with silk-fibroin-like structures [38], it is reasonable to hypothesize that SGF1 plays an important role in activating byssal protein gene expression, which needs further investigation.

### 2.4. AMPK-Interacting Protein SGF1 Is Responsible for Larval Settlement

To explore the potential role of the SGF1 in the larval settlement of *M. sallei*, the expression profile of SGF1 in different larval developmental stages were analyzed by qRT-PCR. The expression of *SGF1* increased gradually during larval development, and reached a maximum in the settled larvae, then fell significantly in the metamorphosed larvae (Figure 5B). Furthermore, the settled larvae of *M. sallei* were immunostained with polyclonal antibodies against SGF1 to determine the tissue localization during larval settlement. As shown in Figure 5C, SGF1 was highly expressed in the foot, with lower expression in the digestive gland. Similar expression patterns were also observed for the AMPK [20]. The foot is an important structure for locomotion in molluscan larvae; in particular, the foot is important for seeking optimal locations for settlement [39,40]. These results suggest that SGF1 does play an important role in the regulation of larval settlement in *M. sallei*.

Moreover, lipofectamine-mediated siRNA transfection was applied for confirming the function of SGF1 in larval settlement. To transfect siRNA into the larvae, they were fed with siRNA at the pediveliger stage. The concentrations of lipofectin and siRNA were set as 2 μL mL^−1^ and 10 μg mL^−1^, respectively. After 12 h of treatment, lipofectin effectively brought siRNA into the larvae, as indicated by the red fluorescence (Figure 6A). In the solvent control, no obvious red fluorescent signal was detected, except for the digestive gland (which showed red autofluorescence). In the treatments, the red fluorescence was widely spread throughout the whole bodies. There was no visual difference in the signal distribution or intensity between the sense RNA and nonsense RNA treatments. As shown in Figure 6B, the transfection of SGF1-2 siRNA significantly inhibited the larval settlement. No effect on the larval settlement was observed for the SGF1-1 siRNA, and the nonsense siRNAs compared with the solvent control. Further qRT-PCR examination revealed that SGF1-2 siRNA transfection successfully suppressed the mRNA levels of SGF1 (Figure 6C). These results suggest that the downregulation of the mRNA levels of SGF1 led to decreased levels of larval settlement, further confirming the importance of SGF1 in regulating the larval settlement of *M. sallei*.

### 2.5. TRPM7-AMPK Pathway Is Involved in Larval Responses to Substrates with Different Stiffness

Larval settlement can be influenced by the surface mechanical properties. Stiffness is one important mechanical property of the surfaces. It has been found that surfaces with higher stiffness accumulate more larvae such as the barnacle *B. neritina*, the polychaete *Polydora ligni*, and the ciliates *Zoothamnium* sp., *Astylozoon* sp., and *Folliculina* sp. [5]. Here, we investigated the larval responses to substrates with different stiffness. The numbers of settled larvae on the relatively stiff PDMS substrates (Young’s modulus were 0.76 ± 0.06 MPa and 0.54 ± 0.10 MPa for 5:1 PDMS and 10:1 PDMS, respectively) were significantly higher than that on the relatively soft PDMS substrate (Young’s modulus was 0.01 MPa for 40:1 PDMS) (Figure 7A,B), suggesting that the larvae preferentially settle on the stiff surfaces. In addition, we observed that the larvae feet crawled more actively on the relatively stiff substrates.

To verify that the TRPM7-AMPK pathway is involved in sensing substrates during the settlement of *M. sallei* larvae, the gene expression change in response to substrates with different stiffness was examined. After a 24 h culture, the gene expression levels of the settled on the substrates decreased gradually as the stiffness decreased (Figure 7B), which showed a close correlation with the number of settled larvae. Overall, the substrate-dependent experiment showed that the mussel larvae were more prone to settle on substrates with high stiffness, on which *TRPM7*, *CaMKKβ*, *AMPK*, and *SGF1* were highly expressed, suggesting that the TRPM7-AMPK pathway is involved in sensing the substrates during larval settlement.

## 3. Materials and Methods

### 3.1. Larval Culture of M. sallei

*M. sallei* adults were collected from the west waters of the Bachimen Seawall of Dongshan, China (23°47′ N, 117°25′ E). Spawning induction and larval culture were carried out in the laboratory following our previously published protocol [41]. Briefly, embryos and larvae were incubated at a density of 3–5 individuals mL^−1^ in filtered seawater (FSW) at 27 ± 1 °C. Larvae were fed with *Dicrateria zhanjiangensis* (Chrysophyta) at a concentration of 1.0–5 × 10^4^ cells mL^−1^. The water was aerated gently and changed daily.

### 3.2. Measurement of Ca^2+^ Flux

To determine whether the TRPM7 channel triggered larval settlement through the Ca^2+^ signal, the Ca^2+^ flux was measured before and after larvae exposure to 10 μM sertraline or mibefradil. Non-invasive micro-test technology (NMT, Xuyue (Beijing, China)) was employed to measure the concentration gradient of Ca^2+^ between the two predetermined points by using a corresponding microsensor (Figure 1A). The Ca^2+^ microsensor was calibrated in 1.0 mM and 0.1 mM Ca^2+^ beforehand. The early settled larvae of *M. sallei* were washed once with the testing liquid (360 mM NaCl, 2.0 mM NaHCO_3_, 8.0 mM KCl, 0.1 mM Na_2_SO_4_, 0.5 mM CaCl_2_, pH 8.1). The testing samples were measured in the measuring chamber containing 3 mL testing liquid with or without TRPM7 channel activators (10 μM), and a Ca^2+^ flux microsensor was positioned near the edge of the shell (approximately 3 μm), before the Ca^2+^ signals were detected. Ca^2+^ flux data were exported from imFluxes V2.0 software.

### 3.3. Genes Expression Analysis by qRT-PCR

For the different developmental stages of *M. sallei* larvae (i.e., umbonate larvae, pediveligers, settled larvae and metamorphosed larvae), samples were collected and frozen in liquid nitrogen, and stored at −80 °C until processing.

For the treatments of the TRPM7 channel activators, approximately 3000 pediveliger larvae were collected and transferred into a 250-mL beak containing 200 mL filtered seawater. Then, the TRPM7 channel activators were added to achieve a concentration of 1 μM to induce larval settlement. After 12 h of culture at 27 °C in the dark, about 1000 settled larvae of each beak were collected and stored at −80 °C until processing. The larvae without exposure to the activators were collected as control.

According to the ORF sequences of *TRPM7*, *CaMKKβ*, *AMPK*, and *SGF1* (Appendix A), matching oligonucleotide primers (Appendix A) were designed. The qRT-PCR analysis was carried out following our published protocol (20). The 2^−∆∆Ct^ Method [42] was utilized to evaluate the relative expression level of genes with β-actin as the internal control.

### 3.4. Screening and Identification of Proteins Interacted with AMPK by Y2H Assay

Details of the construction of the cDNA expression library and bait plasmid are provided in the Appendix A. To screen out proteins that interacted with AMPK, the Yeastmaker™ Yeast Transformation System was adopted to co-transform the prey plasmids and pGBKT7-AMPK. The co-transformants were planted on SD/-Trp-Leu-His + 20 mM 3AT agar plates at 30 °C for 3–7 days. Colonies with diameter >2 mm were selected as the primary interacting proteins for further analysis and then retransferred into the SD/-Trp-Leu-His liquid medium culturing for 2 days to isolate vectors using the Yeast Plasmid Extraction Kit (Solarbio, China). The vectors were identified by PCR using the pGADT7-F/R primers (TAATACGACTCACTATAGG/GGCAAAACGATGTATAAATGA), and analyzed with the online BLAST tool from the NCBI (https://blast.ncbi.nlm.nih.gov, accessed on 1 August 2022) and transcriptome database of *M. sallei*.

### 3.5. Tissue Localization of SGF1 Analysis by Immunofluorescence

Immunofluorescence was carried out as the published protocol [20]. Briefly, the settled larvae of *M. sallei* were labeled with the polyclonal anti-SGF1 primary antibody (Genscript, dilution 1:500) at 4 °C for 6 h. After thorough rinsing, larvae were then labeled with Alexa Fluor 594 goat anti-rabbit IgG antibody (Signalway Antibody, dilution 1:1000) overnight at 4 °C. Stained larvae were washed thoroughly and then observed under a fluorescence microscope. Larvae without labeling the primary antibody were used as the negative control.

### 3.6. Knockdown of TRPM7 and SGF1 Expression by siRNA Interference

The siRNA sequences were synthesized by Zoonbio Biotech (Nanjing, China) Co., Ltd. The nonsense siRNA sequences (negative control, NC) were synthesized simultaneously and compared with our transcriptome database of *M. sallei*; no hits were found. The list of siRNA sequences is shown in Appendix A. To enable visual observation of the transfection process, the siRNAs of SGF1 were labeled with red fluorescent dye Cy5 at the 5′ ends.

RNA interference was conducted following our previous study with some modifications [20]. The concentration of lipofectin was set at 2 μL mL^−1^, and the concentration of siRNA was set at 10 μg mL^−1^. For a 3 mL culture volume, 30 μg siRNA and 6 μL lipofectin were diluted in 0.3 mL DEPC water and then mixed by vortexing. The mixture was incubated at room temperature for 20 min to form siRNA–liposome complexes. Then, the mixture was added to 2.7 mL FSW containing about 2000 pediveliger larvae. The solvent control (SC) was prepared by using the same quantity of lipofectin only. After treatment for 12 h, about 100 larvae were collected for the settlement bioassay [20], and the remaining larvae were collected to detect the expression levels of *TRPM7*, *CaMKKβ*, *AMPK*, and *SGF1* by qRT-PCR as described above.

### 3.7. Preparation of Substrates with Various Stiffness

A PDMS formulation (SYLGARD 184) composed of a silicon elastomer base and a curing agent was purchased from Dow Corning (MI, USA). To prepare PDMS substrates with different stiffness, the silicone elastomer base (poly(dimethyl-methylvinylsiloxane)) and the cross-linker (poly(dimethyl-methylhydrogenosiloxane)) were mixed at four different weight ratios (5:1, 10:1, 20:1, and 40:1). After degassing under centrifugation for 10 min at 8000 rpm, 10 mL of the PDMS formulations was poured into the 250-mL glass beak to investigate larval settlement and *TRPM7* expression difference in response to different substrates.

### 3.8. Larval Settlement and Gene Expression in Response to Different Substrates

FSW (100 mL) containing about 3000 pediveliger larvae were added into the 250-mL glass beaks with different stiffness substrates. After a 24 h culture at 27 °C in the dark, the number of settled larvae on the different substrates was counted through the inverted microscope. Then, the settled larvae were collected to detect the expression levels of *TRPM7*, *CaMKKβ*, *AMPK*, and *SGF1* by qRT-PCR, as described above. Finally, the PDMS films were peeled from the bottom of glass beakers and then cut to test the Young’s modulus according to the Chinese standard, GB/T 13022-1991.

### 3.9. Statistical Analysis

Results were analyzed with SPSS 22.0 software. For the gene expression analysis of the siRNA interferences, and the bioassay of the larval settlement responses to siRNA, one-way analysis of variance (ANOVA) was performed with a Dunnett’s post-hoc test for multiple comparisons of treatment means with the control. For the expression analysis of genes during larval development and the larval responses to different substrates, ANOVA was performed with Tukey’s test for multiple comparisons of treatment means. For the gene expression analysis response to TRPM7 activators, the Student’s *t*-test was used to evaluate the statistical significance between two datasets. All of the significance levels were set at *p* < 0.05.

## 4. Conclusions

Our studies have shown that the TRPM7-mediated Ca^2+^ signal is involved in triggering the larval settlement of *M. sallei* through the CaMKKβ-AMPK-SGF1 pathway. It was found that *M. sallei* larvae preferred the stiff surfaces for settlement, on which *TRPM7*, *CaMKKβ*, *AMPK*, and *SGF1* were highly expressed. These findings will help us to better understand the molecular mechanisms of larval settlement in marine invertebrates. Furthermore, the screening of inhibitors targeting the TRPM7-AMPK pathway as antifouling agents will be a potential direction for developing environmentally friendly antifouling coatings for mussel fouling.

## Figures and Tables

**Figure 1 ijms-24-05399-f001:**
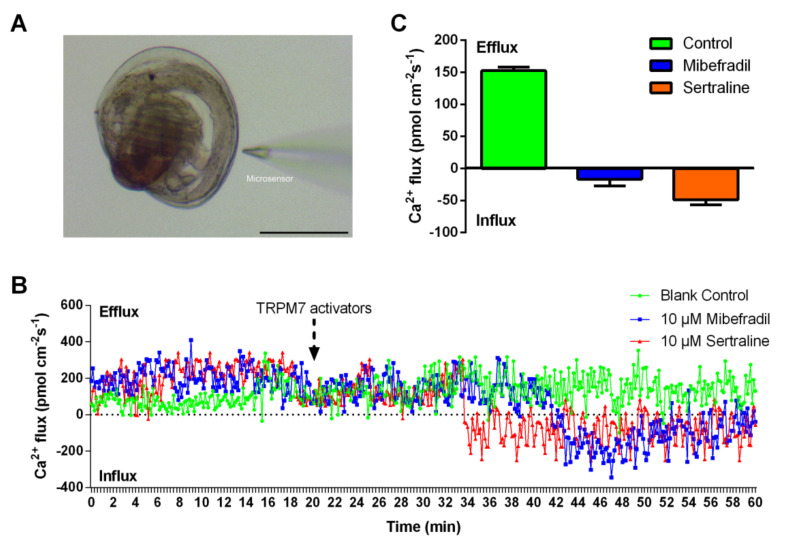
Transient receptor potential melastatin-subfamily member 7 (TRPM7) channel activators induce calcium influx in the larvae of *M. sallei*. (**A**) Image of the NMT test for an early settled larva of *M. sallei*. Scale bar: 100 μm. (**B**) Real-time curves of net Ca^2+^ fluxes of representative larvae of *M. sallei*. (**C**) Integrated Ca^2+^ fluxes over 30 min (corresponding 30−60 min in Figure 1B) after the addition of TRPM7 channel activators (*n* = 6; error bars represent mean ± s.e.m.).

**Figure 2 ijms-24-05399-f002:**
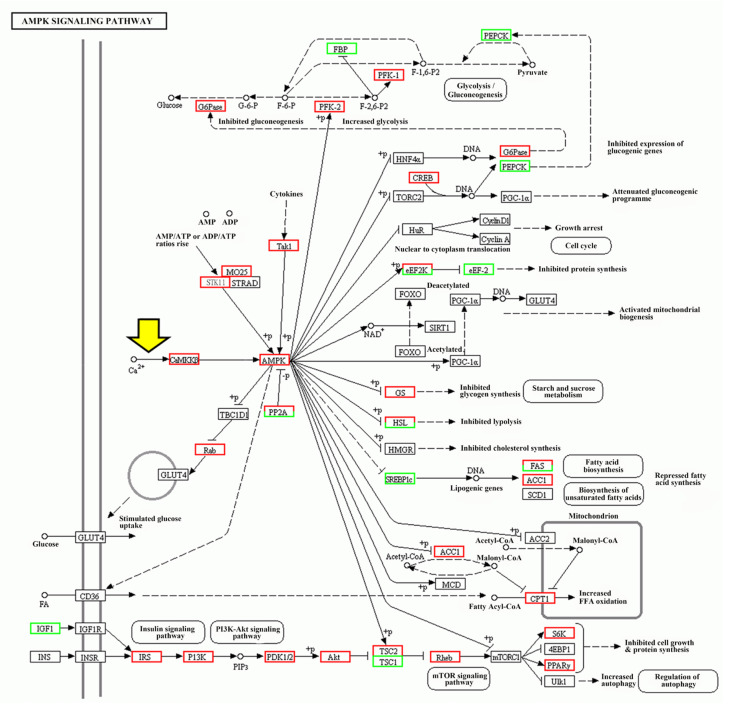
AMP-activated protein kinase (AMPK) signaling pathway (ko01522) constructed based on the KEGG pathway analysis for settled larvae vs. pediveliger larvae. Red and green boxes represent the genes that were upregulated and downregulated, respectively. Mixed colored boxes indicate that there were several genes annotated including upregulated (red) and downregulated genes (green). The thick yellow arrow indicates that intracellular Ca^2+^ can activate the CaMKKβ-AMPK pathway.

**Figure 3 ijms-24-05399-f003:**
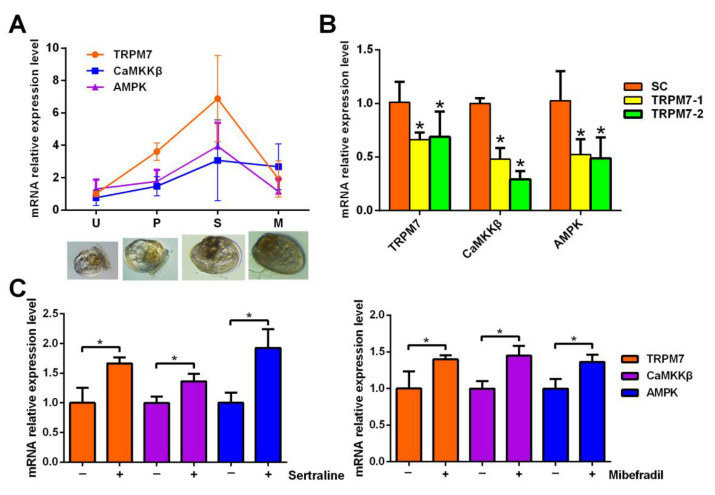
Genes expression of the TRPM7 channel associated with CaMKKβ/AMPK at the mRNA level. (**A**) Relative expression levels of *TRPM7*, *CaMKKβ*, and *AMPK* during larval development. U: umbone larvae. P: pediveligers; S: settled larvae; metamorphosed larvae. (**B**) Relative expression level of *TRPM7* associated with *CaMKKβ*/*AMPK* after larvae exposure to siRNA against *TRPM7* for 12 h. SC: solution control, 2 μL mL^−1^ lipofectin. * Denotes a significant difference between the treatments and the control (*p* < 0.05, Dunnett’s test). (**C**) Relative expression level of *TRPM7* associated with *CaMKKβ*/*AMPK* after larvae exposure to the TRPM7 channel activators for 12 h. * Denotes a significant difference between the treatments (*p* < 0.05, Student’s *t*-test). Data are indicated as the mean ± SD from triplicate experiments.

**Figure 4 ijms-24-05399-f004:**
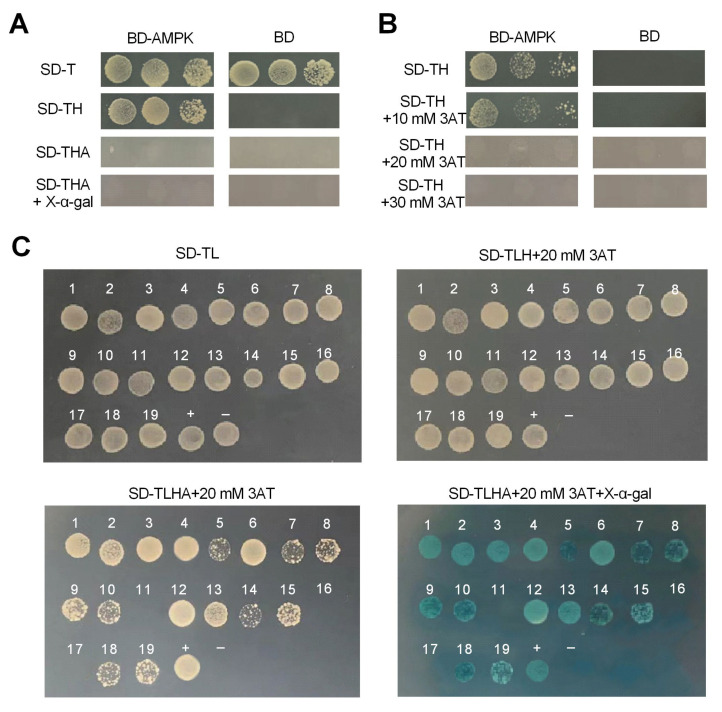
Screening of proteins interacting with AMPK. (**A**) Auto-activation assay of the bait plasmid pGBKT7−AMPK. (**B**) 3-Aminotriazole (3AT) was used to inhibit the auto-activation of the bait plasmid. (**C**) Yeast-two-hybrid screening of the interactions between the bait plasmid and prey plasmids. Positive control: co-transformants containing pGADT7−T and pGBKT7-p53; Negative control: co-transformants containing pGADT7−T and pGBKT7−laminC.

**Figure 5 ijms-24-05399-f005:**
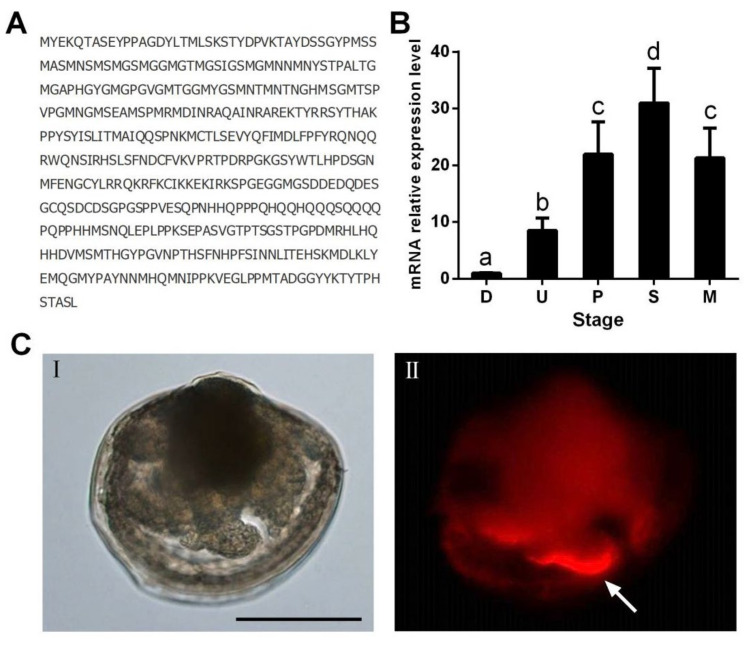
Temporal and spatial expression patterns of Silk gland factor 1 (SGF1). (**A**) The deduced amino acid sequence of SGF1. (**B**) Expression of SGF1 during larval development at the mRNA level. Different letters above the bars denote significant differences among treatments (*p* < 0.05, Tukey’s test). D: D-shaped veligers; U: umbone larvae; P: pediveligers; S: settled larvae; M: metamorphosed larvae. (**C**) Tissue localization of SGF1 in settled larva. I is the bright field picture; II is the fluorescent field picture of I. Arrow indicates foot. Scale bar: 100 μm.

**Figure 6 ijms-24-05399-f006:**
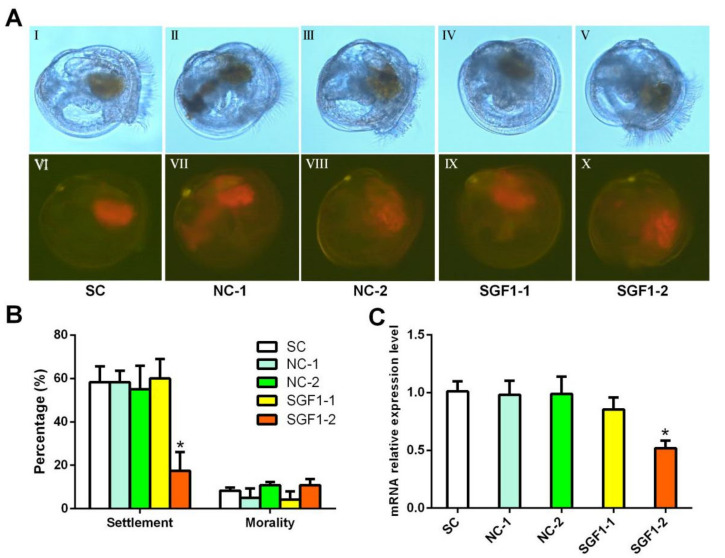
siRNA transfection against SGF1 inhibited the larval settlement of *M. sallei*. (**A**) Pediveliger larvae were treated for 12 h with 10 μg mL^−1^ 5′cy5-labeled siRNA against SGF1. I, II, III, IV, and V are the bright field pictures; VI, VII, VIII, IX, and X are the fluorescent field pictures of I, II, III, IV, and V, respectively. (**B**) Larval settlement and mortality of *M. sallei* after siRNA transfection against SGF1. (**C**) Relative expression level of SGF1 mRNA after siRNA interference. SC: solvent control, 2 μL mL^−1^ lipofectin; NC-1: negative control, the nonsense siRNA-1; SGF1-1: the sense siRNA-1; NC-2: negative control, the nonsense siRNA-2; SGF1-2: the sense siRNA-2. * Denotes a significant difference between the treatments and the solvent control (*p* < 0.05, Dunnett’s test).

**Figure 7 ijms-24-05399-f007:**
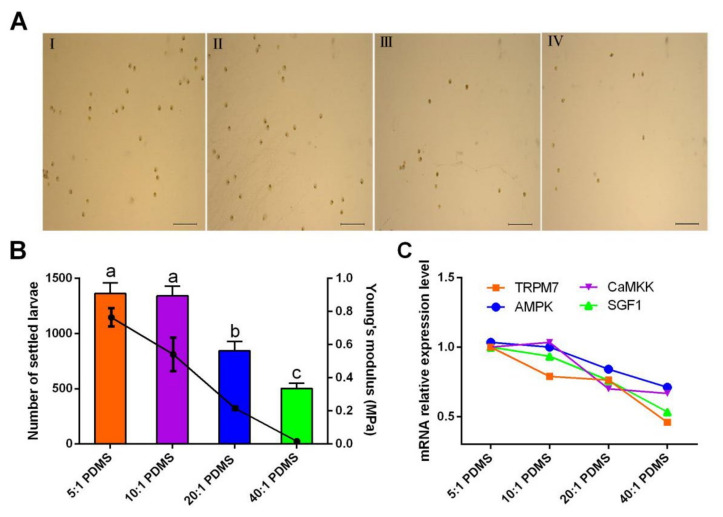
Larval responses to substrates with different stiffness. (**A**) Micrographs show the larval settlement on different PDMS substrates. I, II, III, and IV are the 5:1, 10:1, 20:1, and 40:1 PDMS substrates, respectively. Scale bar: 2 mm. (**B**) Number of settled larvae on different PDMS substrates. The line graph indicates the stiffness of different PDMS substrates expressed by the Young’s modulus. Different letters above the bars denote significant differences among treatments (*p* < 0.05, Tukey’s test). (**C**) Gene expression difference in response to different PDMS substrates. 5:1, 10:1, 20:1, and 40:1 PDMS indicate the weight ratios of the silicone elastomer base and the cross-linker in the PDMS substrates.

**Table 1 ijms-24-05399-t001:** The analysis results of the prey proteins blasted against the NCBI database and the transcriptome database of *M. sallei*.

No.	Protein Name	Accession	No.	Protein Name	Accession
1	Mitochondrial ATP synthase lipid-binding protein-like protein (ATPS)	MZ592789.1	9	Ferritin	JQ959539.1
2	E3 ubiquitin-protein ligase TRIM56 (E3UPL)	XM_045315934.1	10	Tubulin beta-1 chain (Tubulin β-1)	XM_003979430.2
3	Elongation factor 2 (EF-2)	XM_012821713.1	12	6-phosphofructo-2-kinase (PFK-2)	XM_011454180.2
4	60S acidic ribosomal protein P2 (RPP2)	XM_005102789.3	13	Adenine nucleotide translocator (ANT)	KX361239.1
5	Silk gland factor 1-like (SGF1)	XP_021361791.1	14	Calcium/calmodulin-dependent protein kinase kinase 2 (CaMKKβ)	XR_001204123.1
6	Cytochrome P450 4B1 (P450)	XM_016429472.1	15	Serine/threonine kinase 11 (STK11)	XM_012645487.1
7	Glycogen synthase (GS)	XM_014826310.1	18	Acetyl-CoA carboxylase (ACACA)	XM_014254957.1
8	Uncharacterized protein	_	19	Uncharacterized protein	_

## Data Availability

All of the data used in this study have been provided in the main text and the Appendix A.

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
