# Peer review of "TRPM7-Mediated Ca2+ Regulates Mussel Settlement through the CaMKKβ-AMPK-SGF1 Pathway"

_ijms, 2023, doi:10.3390/ijms24065399_

Round 1

Reviewer 1 Report

This paper describes a very important problem, identifying the signaling  pathways responsible for modulating settlement of larvae on different substrates. However, in its current form the paper is not written in sufficient detail that the results can be verified. 
The target organism is M. Sallei but sequences and databases and gene names provided are from a variety of other organisms without any justification what the sequence identities are and how the best matches were chosen. For example the Kegg image is simply downloaded from keg without adaptation to this organism. The table that provides lists of RNA’s studied has “uncharacterized protein” as multiple entries - which is impossible to verify what the authors were using or looking at.

for assays to write “unpublished results” as reference is also not useful if we don’t know what is being referenced.

the proteins for the yeast two hybrid assay also appeared out of the blue. how are they connected to trp channels or chosen?

the clone numbers in the graph are not informative without providing a list of what each clone number referred to.

Often background references are provided after the description of results although the information should have been provided in introduction.

the results are not placed in context of how they compare to other organisms, at the mechanistic level.

how do we know that the reagents used, eg activators and inhibitors of the channel work, how do the magnitude of the traces compare to model organisms?

Reviewer 2 Report

This manuscript titled “TRPM7-mediated Ca 2+ regulates mussel settlement through the 2 CaMKKβ-AMPK-SGF1 pathway”. The comments for this manuscript are as follows:

The research level of the whole manuscript is good, but whether it is possible to specifically write in the section of conclusions how to study these mechanisms for the future application of developing friendly environment and antifouling coatings for fouling organisms. There can be more narratives, which will be of greater significance to readers. Otherwise, this research may only be reduced to a basic research that is difficult to apply.

Author Response

Comments and Suggestions for Authors

This manuscript titled “TRPM7-mediated Ca2+ regulates mussel settlement through the CaMKKβ-AMPK-SGF1 pathway”. The comments for this manuscript are as follows:

The research level of the whole manuscript is good, but whether it is possible to specifically write in the section of conclusions how to study these mechanisms for the future application of developing friendly environment and antifouling coatings for fouling organisms. There can be more narratives, which will be of greater significance to readers. Otherwise, this research may only be reduced to a basic research that is difficult to apply.

Response: We thank the reviewer for the positive comments. As suggested, we have added the narratives on how to study these mechanisms for the future application of developing environmentally friendly antifouling coatings to the conclusions (lines 422-424).

Round 2

Reviewer 1 Report

I find that the authors addressed my concerns. Congratulations on a very comprehensive study!